# An Application of Machine Learning That Uses the Magnetic Resonance Imaging Metric, Mean Apparent Diffusion Coefficient, to Differentiate between the Histological Types of Ovarian Cancer

**DOI:** 10.3390/jcm11010229

**Published:** 2021-12-31

**Authors:** Heekyoung Song, Seongeun Bak, Imhyeon Kim, Jae Yeon Woo, Eui Jin Cho, Youn Jin Choi, Sung Eun Rha, Shin Ah Oh, Seo Yeon Youn, Sung Jong Lee

**Affiliations:** 1Department of Obstetrics and Gynecology, Seoul St. Mary’s Hospital, College of Medicine, The Catholic University of Korea, 222, Banpo-daero, Seocho-gu, Seoul 06591, Korea; songdeng77@naver.com (H.S.); newsseong@gmail.com (S.B.); kih1119@naver.com (I.K.); woo.jaeyeon@gmail.com (J.Y.W.); twikkling@naver.com (E.J.C.); yunno@catholic.ac.kr (Y.J.C.); 2Department of Radiology, Seoul St. Mary’s Hospital, College of Medicine, The Catholic University of Korea, 222, Banpo-daero, Seocho-gu, Seoul 06591, Korea; serha@catholic.ac.kr; 3NAVER Clova, 246, Hwangsaeul-ro, Bundang-gu, Seongnam-si 13595, Korea; so2461@columbia.edu

**Keywords:** ovarian epithelial cancer, magnetic resonance imaging, apparent diffusion coefficient, machine learning

## Abstract

This retrospective single-center study included patients diagnosed with epithelial ovarian cancer (EOC) using preoperative pelvic magnetic resonance imaging (MRI). The apparent diffusion coefficient (ADC) of the axial MRI maps that included the largest solid portion of the ovarian mass was analysed. The mean ADC values (ADC_mean_) were derived from the regions of interest (ROIs) of each largest solid portion. Logistic regression and three types of machine learning (ML) applications were used to analyse the ADCs and clinical factors. Of the 200 patients, 103 had high-grade serous ovarian cancer (HGSOC), and 97 had non-HGSOC (endometrioid carcinoma, clear cell carcinoma, mucinous carcinoma, and low-grade serous ovarian cancer). The median ADC_mean_ of patients with HGSOC was significantly lower than that of patients without HGSOCs. Low ADC_mean_ and CA 19-9 levels were independent predictors for HGSOC over non-HGSOC. Compared to stage I disease, stage III disease was associated with HGSOC. Gradient boosting machine and extreme gradient boosting machine showed the highest accuracy in distinguishing between the histological findings of HGSOC versus non-HGSOC and between the five histological types of EOC. In conclusion, ADC_mean_, disease stage at diagnosis, and CA 19-9 level were significant factors for differentiating between EOC histological types.

## 1. Introduction

Ovarian cancer is the eighth most common cancer in women, with 313,959 new diagnoses worldwide in 2020 [1]. High-grade serous ovarian cancer (HGSOC) accounts for up to 80% of epithelial ovarian carcinomas (EOCs) [1,2,3]. HGSOCs are found at more advanced stages with multiple metastases and are characterised by P53 mutations with frequent genomic instability due to defects in the pathways contributing to DNA repair [4,5]. On the other hand, in non-HGSOC, mucinous carcinoma cancer is frequently diagnosed at an early stage [6]. However, mucinous carcinoma is more difficult to surgically excise in advanced ovarian cancer than other histologic types of epithelial ovarian cancer [7,8]. Clear-cell and mucinous carcinomas are characterised by resistance to carboplatin and paclitaxel chemotherapy [9,10,11]. Therefore, the preoperative prediction of histological types could be useful when counselling patients regarding fertility-sparing surgery, extent of cytoreductive surgery, decision to undergo neoadjuvant chemotherapy, and survival outcomes.

Magnetic resonance imaging (MRI), which is the preferred imaging modality for ovarian cancer owing to its ability to provide high contrast for the visualisation of soft tissues, has been utilised for the differential diagnosis of HGSOC [12]. HGSOC generally presents as a smaller ovarian mass, with more nodules frequently seeding the peritoneum and more frequent solid cystic tumours than non-HGSOC [13]. However, morphological interpretations are not sufficiently accurate to differentiate between the different histological types of EOC. This is attributed to the very similar morphological characteristics of the EOC types. Diffusion-weighted imaging (DWI) is a unique MRI technique based on the impedance of water diffusivity [14]. It has improved the accuracy of detection and characterisation of malignant tumours in gynaecologic cancer [15]. A quantitative metric, the apparent diffusion coefficient (ADC), was assessed using DWI. The ADC is helpful in evaluating whether a tumour is malignant and allows prognostic assessment of the tumour [14,16,17,18].

Machine learning (ML) was recently introduced to assist in providing an accurate analysis of clinical findings and making treatment decisions [19,20]. Furthermore, the use of ML has been expanded to include assistance in the detection of abnormal lesions or analysis of the characteristics of various images [21]. In this study of patients with EOC, we used ML to perform a preoperative analysis of the impedance imaging characteristics of MR images and the clinical factors of patients to predict the histological types of EOCs.

## 2. Methods

### 2.1. Study Participants and Clinical Findings

This retrospective, single-centre study included patients who underwent preoperative pelvic MRI between January 2009 and December 2020 and received a final histopathological diagnosis of EOC from the excised specimen. Patients who were diagnosed with concomitant primary cancers, whose MRIs were not applicable for DWI and ADC mapping, or who did not have a measurable ovarian mass suitable for a confirmatory diagnosis were excluded (Figure 1). The following data were retrospectively obtained for analysis: clinical factors such as patient age, levels of tumour markers (CA 125, CA 19-9); MRI findings that included the mean ADC value of the solid portion, International Federation of Gynecology and Obstetrics (FIGO) stage at diagnosis, and histological features of the excised specimen.

### 2.2. MRI Protocol and Interpretation

MR images were obtained using 3.0 T MRI systems (Verio, Siemens Healthcare, Erlangen, Germany; Vida, Siemens Healthcare, Erlangen, Germany). Conventional pelvic MRI included T2-weighted half-Fourier single-shot turbo spin echo, T2-weighted turbo spin echo, T1-weighted turbo spin echo, and contrast-enhanced T1-weighted turbo spin echo images. DWIs with b-values of 0 and 800 s/mm^2^ were obtained, and ADC maps were constructed from these DWIs.

A gynaecologist with six years of experience in gynaecology and a radiologist with seven years of experience in gynaecological MRI reviewed the pelvic MRIs. All reviewers were blinded to the histological findings. After reviewing the entire sequence of each patient’s MRI scan, the largest axial ADC map of the solid portion of the ovarian cancer was selected to delineate the region of interest (ROI). The ROI in the selected solid portion was manually drawn using the picture archiving and communication system (PACS) (Figure 2). The ADC_mean_ value of each solid portion was derived from the ROI in the PACS. An ROI encompassing the entire ovarian lesion was drawn on the same axial image. First, the ROIs of the solid portion and the entire ovarian mass were independently drawn by a gynaecologist and radiologist. The final ROIs were then selected after a consensus meeting by the two reviewers. The two reviewers together then measured the longest diameters of the entire ovarian mass and the solid portion, identified the location of the ovarian lesion, and assessed the presence of radiologic peritoneal seeding.

### 2.3. Statistical Analysis

The clinical and radiological features of patients with HGSOC and patients with non-HGSOC were compared using the Mann–Whitney U test for nonparametric continuous variables and the chi-squared test for categorical variables. Statistical significance was set at *p* < 0.05.

The Kruskal–Wallis test was then performed to assess the differences between the parameters of patients with HGSOC, clear-cell carcinoma (CCC), mucinous carcinoma (MC), endometrioid carcinoma (EC), and low-grade serous ovarian carcinoma (LGSOC). The Mann–Whitney U test and Bonferroni’s method were used for post-hoc analysis of the groups of patients stratified by specific tumours to identify significant differences. A logistic regression model was used to compare patients with HGSOC to those with non-HGSOC to evaluate the predictability of the patients’ MRI findings and their clinical features for histological types of EOC. Differences between factors that were found to be statistically significant by the Mann–Whitney U test were evaluated using logistic regression analysis.

A receiver operating characteristic (ROC) curve was generated for patients with HGSOC versus those without HGSOC. The optimal cut-off value, sensitivity, specificity, positive predictive value, and negative predictive value were determined at the point at which the Youden index (sensitivity + specificity −1) was maximal. The area under the curve (AUC) was calculated as follows: the reliability of the ADC_mean_ values between gynaecologists and radiologists was evaluated using the intraclass correlation coefficient (ICC). SPSS software (version 24; SPSS Inc., Chicago, IL, USA) was used for statistical analysis.

Random forest, gradient boosting machines (GBMs), and extreme gradient boosting (XGBoost) are machine learning (ML) tools used in this study. The algorithms were chosen after initial experiments using the AutoML method, through which a wide range of models were automatically selected, trained, and tuned. The hyperparameters were tuned using the grid search method over tuneable hyperparameters for each algorithm. To calculate representative performance metrics for each model, five trainings, each using 5-fold cross-validation, were performed. Variable importance was extracted, and heatmaps were constructed to compare the effects of each variable on the outcomes of the different types of models. All machine learning procedures were performed in Python and using the H2O.ai API. The mean value of the 5-fold cross validation of the gradient boosting machine tool was used to generate ROC curves. The accuracy was derived from the ratio of the number of correct predictions to the total number of input samples.

## 3. Results

### 3.1. Clinical Factors of the Study Population

Of the 200 enrolled study patients, 103 had high-grade serous ovarian cancer (HGSOC, *n* = 103) and 97 had non-HGSOC, consisting of CCC (*n* = 35), MC (*n* = 33), EC (*n* = 21), and LGSOC (*n* = 8). Borderline ovarian tumours (BOTs) were excluded from the original analysis since five of 12 (41.7%) patients with borderline tumours had no measurable solid portion on preoperative MRI. The median age of the patients in the HGSOC group was significantly higher than that of the patients in the non-HGSOC group (median age: 55 years vs. 50 years, respectively; *p* < 0.001). Most patients with HGSOC had stage III disease at the time of diagnosis (*n* = 58, 56.3%) and most patients with non-HGSOC had stage I disease at the time of diagnosis (*n* = 65, 67.0%, *p* < 0.001). The level of CA 125 and prevalence of peritoneal seeding were significantly higher in patients with HGSOC than in those without HGSOC (median CA 125: 671.8 IU/mL vs. 58.8 IU/mL, respectively; *p* < 0.001). The CA 19-9 and the diameter of the entire ovarian lesion were lower in patients with HGSOC than in those without HGSOC. The median value of the ADC_mean_ was lower in the patients with HGSOC than in the patients with non-HGSOC (median value of ADC_mean_: 1.05 × 10^−3^ s/m^2^ vs. 1.53 ×10^−3^ s/m^2^, respectively; *p* < 0.001) (Table 1).

### 3.2. ADC_mean_ Values and Proportions of the Solid Portion in Each Histological Type of EOC

The median value of the ADC_mean_ was significantly lower for the patients with HGSOC, at 1.05 × 10^−3^ s/m^2^, compared with the values for the other histological types (CCC, MC, and EC) (respective median values of the ADC_mean_ of CCC, MC, and EC = 1.62 × 10^−3^ s/m^2^, 1.60 × 10^−3^ s/m^2^, and 1.27 × 10^−3^ s/m^2^; all *p* < 0.001). However, the median value of the ADC_mean_ of patients with HGSOC was lower than that of patients with LGSOC (1.36 × 10^−3^ s/m^2^), but the difference was not significant. Additionally, among patients with non-HGSOC, the ADC_mean_ value of patients with EC was the lowest and increased in the order of LGSOC, MC, and CCC. The only significant difference was noted between EC and MC (*p* < 0.001) (Figure 3).

Regarding the ratios of the area of the solid portion to the total area of ovarian lesions as assessed from the ROIs, the ratio of the area of the solid portion to the total area of ovarian lesions was significantly higher in patients with HGSOC than in those with non-HGSOC (0.49 vs. 0.31, respectively; *p* = 0.001). Among the five histological types of EOC, the ratio of the solid portion to the total ovarian lesion was significantly higher in patients with HGSOC than in those with MC (0.49 vs. 0.17, respectively; *p* < 0.001). In addition, EC had the highest ratio of the area of the solid portion to the total area of ovarian lesions, and the ratios of CCC, LGSOC, and MC decreased in non-HGSOC; however, the difference was not significant (Figure 3).

### 3.3. Subanalysis for Stage I EOC Including BOT

ADC_mean_, CA 19-9, and age at diagnosis were noted to be significantly different between HGSOC and non-HGSOC by univariate analysis (*p* = 0.19, 0.21, and 0.36, respectively). Among the five histological types of EOC, ADC_mean_, CA 19-9, and the ratio of the area of the solid portion to the total area of the ovarian lesions were statistically significant factors. However, ADC_mean_ was a significant factor only between EC and MC (*p* = 0.009), and the ratio of the area of the solid portion to the total area of the ovarian lesions was statistically significant between CCC and LGSOC (*p* < 0.001) in post-hoc analysis. In the confined early stage, a difference among non-HGSOC patients was not clearly observed. However, the ADC_mean_ and ratio of the area of the solid portion to the total area of ovarian lesions had significant roles in distinguishing each histological type of stage I EOC, similar to the ADC_mean_ of all stages of EOC (Appendix A).

### 3.4. Logistic Regression Analysis

Logistic regression analysis was conducted between HGSOC and non-HGSOC. The only factors that were statistically significant in the univariate analysis were included in the logistic regression. After completing the analysis, the age at diagnosis, ADC_mean_, CA 19-9, and stage were selected as meaningful variables in the calculated odds ratio and *p*-value. Lower ADC_mean_ values and CA 19-9 levels were independent predictors significantly related to HGSOC (odds ratios: 0.996 and 0.973, *p* = 0.001, 0.021, respectively). Compared to stage I disease, stage III disease was significantly associated with HGSOC (odds ratio: 5.98, *p* = 0.007) (Table 2). The AUCs of the mean ADC value and the CA 19-9 level were 0.87 and 0.77, respectively. The ADC_mean_ value showed the highest AUC by logistic regression analysis, and detailed findings showed that the cut-off value was 1.26 × 10^−3^ s/m^2^, with sensitivity and specificity as 0.85 and 0.77, respectively, and a positive predictive value and negative predictive value of 0.79 and 0.83, respectively (Figure 4).

The AUCs of the ADC_mean_ value and the GBM were 0.87 and 0.93, respectively. The mean value of the five-fold cross-validation of the GBM tool was used to generate the ROC curves. ROC, receiver operating characteristic; GBM, gradient boosting machine; ADC_mean_, mean value of the apparent diffusion coefficient.

### 3.5. Machine Learning Analysis

GBM was the most accurate among the three types of ML tools that evaluated the MRI parameters and clinical features of patients with HGSOC and non-HGSOC (accuracy: 0.91, AUC: 0.93, area under the precision recall curve: 0.91) (Figure 4). For the five histological types of EOC, XGBoost showed the highest accuracy among the three ML tools (accuracy, 0.68; AUC, 0.83; AUCPR, 0.64) (Table 3). Among patients with HGSOC versus non-HGSOC, the ADC_mean_ and FIGO stages at diagnosis were 34.9% and 25.5%, respectively. Among the five histological types of EOC, the ADC_mean_ value and CA 19-9 level were 24.5% and 12.2%, respectively (Figure 5).

## 4. Discussion

This study demonstrated that the ADC_mean_ values of the solid components were significantly correlated with the histological types of EOC, not only for HGSOCs and non-HGSOCs, but also for the five histological types of EOC (HGSOC, EC, CCC, MC, LGSOC). In previous studies, the morphological characteristics of MR images, tumour markers, signal intensity of DWI, and ADC values were studied to determine correlations with the histological types of EOC [22,23]. Pi et al. conducted a meta-analysis on ADC values in the solid portion of tumours and reported excellent diagnostic performance for the ADC value in distinguishing between benign ovarian tumours and malignant ovarian tumours (AUC 0.96, sensitivity 0.91, specificity 0.91) [22]. Tanaka et al. found that bilateral disease, small tumours, and increased signal intensity on DWI were representative of serous carcinomas, while multilocular cysts and increased CA 19-9 levels were representative of mucinous tumours [23]. Zhang et al. found that an increased mean ADC value was more likely to suggest type I EOC (odds ratio (OR) = 16.80, *p* < 0.01); however, the study was limited by selection bias, as patients with borderline epithelial tumours comprised 51% of the study patients with type I ovarian cancer [24]. Therefore, to the best of our knowledge, ours is the first study to evaluate the correlations between five histological types of EOC, and we believe that the ADC findings correlated with the histology of EOCs are an important contribution to the findings of previous studies.

Among the five histological types of EOCs present in our study, HGSOCs exhibited the lowest mean ADC values. This finding can be attributed to the high cellularity of the HGSOCs. A typical characteristic of HGSOC is its compact structure, which consists of a solid mass of cells with marked nuclear atypia [25]. Since ECs have a similar structure, showing a solid mass of cells, immunohistochemical analysis is crucial for differentiating between the two types of EOC. Our results showed that HGSOC exhibited denser cellularity than EC did [22,24,26,27]. In contrast, the lower cellularity of MC was found to be accompanied by an abundant mucinous component, which resulted in a higher mean ADC value than HGSOC [28].

The relationship between the mean ADC value and cellularity in ECs or CCCs is controversial. Tanaka et al. reported that CCC had a higher proportion of solid components than EC. On the contrary, we found that CCC had a lower proportion of solid components than EC, similar to the results of Ono et al. [23,29]. According to the World Health Organization, CCC is histologically defined as a neoplasm composed of clear cells growing in a solid, tubular, or papillary pattern, with hobnail cells lining the tubules and cysts [30]. However, correct histopathological diagnosis can be challenging due to the rarity of CCCs, which have a wide range of histological features [31]. This histological diversity can be used to derive the difference in the mean ADC values for CCC.

The ratio of the area of the solid portion to the total area of ovarian lesions also differed between the histological types of EOC. The ratio was noted to be significantly different between HGSOC and non-HGSOC, HGSOC, and MC. This result was the same as the result of a previous study that showed that HGSOC had a higher proportion of solid lesions than MC [23]. However, this factor was not useful for distinguishing between each histological type of EOC using logistic regression analysis. ML tools also did not find that the proportion of solid HGSOC was of significant importance (Figure 5). Only a few studies on the ratio of the area of the solid portion to the total area of ovarian lesions have been performed to distinguish between borderline epithelial tumours and EOC, primary fallopian cancer and EOC, or different types of metastatic ovarian cancer [32,33,34]. Therefore, further studies are needed to determine the ratios of the areas of the solid portion to the total area of ovarian lesions.

Compared with patients with HGSOC, those without HGSOC were diagnosed at an early stage. Approximately 60% of CCC or EC cases and 80% of MCs are discovered and histologically diagnosed at stage I [35,36,37]. The therapeutic strategy for non-HGSOC in the early stage was more diverse than that for HGSOC. MC or LGSOC were eligible for fertility-sparing surgery [36], even though early-stage patients diagnosed with CCC may not be indicated for fertility-sparing treatment [10,37]. In addition, systematic lymphadenectomy was controversial in early EOC since systematic lymphadenectomy was increased to detect metastatic lesions but was not associated with improved progression-free survival or OS [35]. Therefore, preoperative histology of EOC is helpful in deciding the treatment option. ADC_mean_, the ratio of the area of the solid portion to the total area of ovarian lesions, and CA 19-9 were related to different non-HGSOC histological types. Considering this, the result of this study is a cornerstone for preoperative histological evaluation. A multicentre study in only stage I EOC will be necessary to obtain a more precise result.

The importance of the ADC_mean_ values, which were obtained through manually drawn ROIs, other MRI characteristics, and clinical findings, was analysed in this study using ML algorithms. A preliminary evaluation used auto ML to select three types of ML. Compared to the conventional logistic regression analysis performed for HGSOC and non-HGSOC, ML provided more accurate results for the accuracy, AUC, and AUCPR values. AUCPR, a useful method for evaluating groups with imbalanced outcomes, was also calculated in this study [38].

Among the patients with HGSOC and non-HGSOC, GBM showed the highest accuracy (0.91), with an AUC of 0.93 and an AUCPR of 0.91. These results were better than those of the conventional logistic regression model of mean ADC value (Figure 4) and similar to those of Qian et al., derived from mixed radiomics models for type I/II EOCs (AUC = 0.96, 95% CI = 0.92–1) [39]. For the five histological types of EOC, the XGBoost ML tool provided the highest accuracy (0.68), with an AUC of 0.83 and AUCPR of 0.64. The XGBoost tool provided a scalable ML system for tree boosting, which is able to solve real-world scaling problems while using a minimal amount of resources [40]. The XGBoost model confirmed the possibility of classifying EOC into five histological types, a result that could not be obtained in previous studies.

Deep learning, a subset of ML, has recently been introduced to extract features from MR images without manually drawing the ROI on an image of the tumour. Although the process is time-consuming, this method provides excellent accuracy [41]. Regarding the accuracy (0.68) for distinguishing between the five histological types of EOC in our study, radiomics combined with the use of deep learning might lead to an improvement in the accuracy of distinguishing between the five histological types of EOC.

Through ML analysis, the ADC_mean_ value was found to be the strongest factor distinguishing between the five histological types of EOC. The percentage importance allowed the identification of the most essential predictor of outcome (importance of ADC_mean_ value: 34.9%; Figure 5). Along with the disease stage at diagnosis, the CA 19-9 level was also meaningful in distinguishing between the five histological types of EOC. Several reports concluded that the CA 19-9 level was predictive only for MC [42,43]. Compared with previous studies, our study expanded on the role of CA 19-9.

Our study was limited in that it was a retrospective, single-centre study. Therefore, there is a possibility of missing clinical data, which may have resulted in bias. Second, only a five-fold validation was performed, instead of external validation. Third, we analysed the impact of the ADC_mean_ value in consensus and not the independent ADC_mean_ value of each reviewer. However, it was found that the ICC between the two reviewers (0.966) was excellent. Fourth, we believe that a larger population will be necessary to incorporate MRI ADC uniformly to advanced stage. Lastly, borderline tumours were not included in this study due to the high percentage (5 of 12, 41.7%) in the non-measurable solid portion on preoperative MRI.

This study has several strengths. First, it was relatively large compared to previous studies. More histological types of EOC were included, which allowed the differentiation of the five histological types of EOC. Second, ML, which is more accurate than conventional analysis, was applied to compensate for several limitations and evaluate the data in this study. Third, the importance of each factor in distinguishing between histological types of EOC was calculated. The ADC_mean_ value, disease stage at diagnosis, and CA 19-9 level were meaningful factors for distinguishing between the histological types of EOC. These results formed the basis for creating a predictive computer program to distinguish between the histological types of EOC.

## 5. Conclusions

This study showed that the ADC_mean_ value, disease stage at diagnosis, and CA 19-9 level were meaningful factors for distinguishing between the histological types of EOC. Moreover, we also found that ML identified the ADC_mean_ value as predictive of histological types and that ML was superior to conventional logistic regression analysis. Preoperative MRI is mandatory for patients with ovarian cancer.

## Figures and Tables

**Figure 1 jcm-11-00229-f001:**
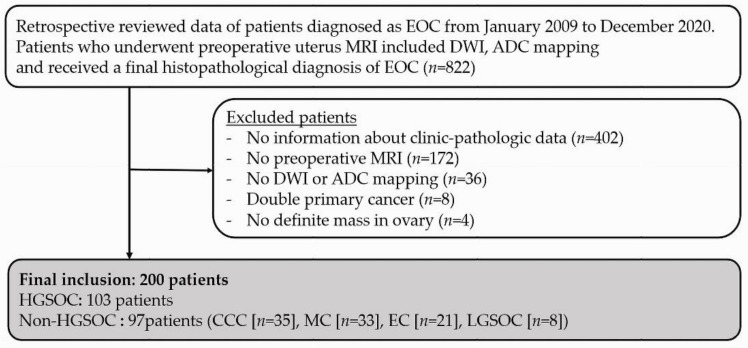
Flowchart for including patients in the study. EOC, epithelial ovarian cancer; DWI, diffusion-weighted imaging; HGSOC, high-grade serous ovarian cancer; MC, mucinous carcinoma; EC, endometrioid cancer; CCC, clear-cell carcinoma; LGSOC, low-grade serous ovarian cancer.

**Figure 2 jcm-11-00229-f002:**
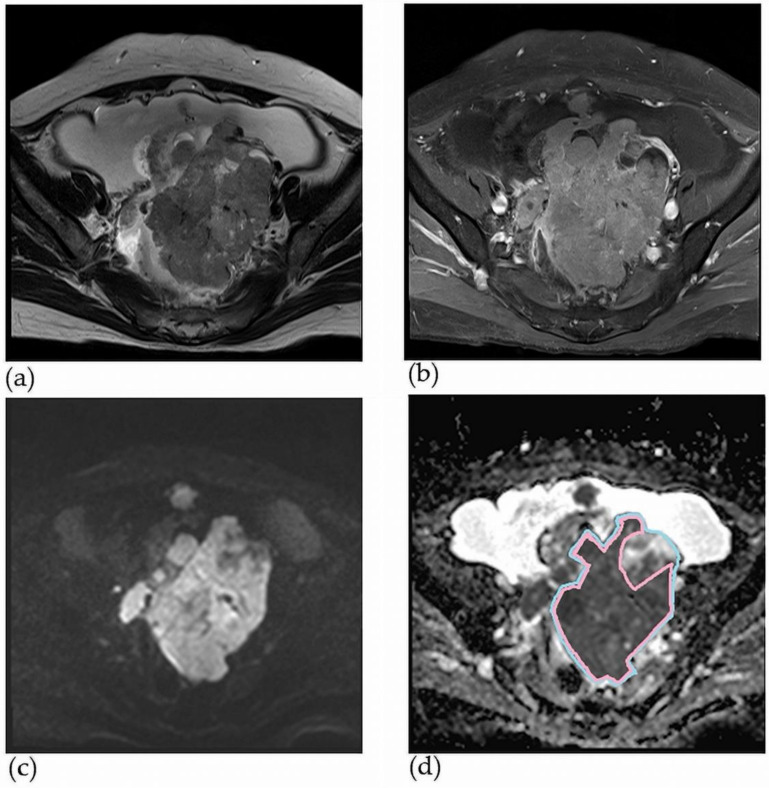
Method for drawing ROI in the solid portion and total ovarian lesion on the MRI of a 65-year-old woman with HGSOC (stage: IIIC), (**a**) the ovarian mass with solid portion on axial T2WI (**b**) the ovarian mass with solid portion on axial contrast-enhance T1WI, (**c**) focal increased signal intensity in the solid component of the mass on DWI (b = 1000 $s/{\mathrm{mm}}^{2}$), (**d**) hypointense solid component on axial ADC map (the mean ADC value is 0.878 × $10^{-3}\;s/m^{2}$ blue line: total ovarian lesion, pink line: solid portion). ROI: region of interest, MRI: magnetic resonance imaging, HGSOC: high-grade serous ovarian cancer, T2WI: T2-weighted image, T1WI: T1-weighted image, ADC: apparent diffusion coefficient.

**Figure 3 jcm-11-00229-f003:**
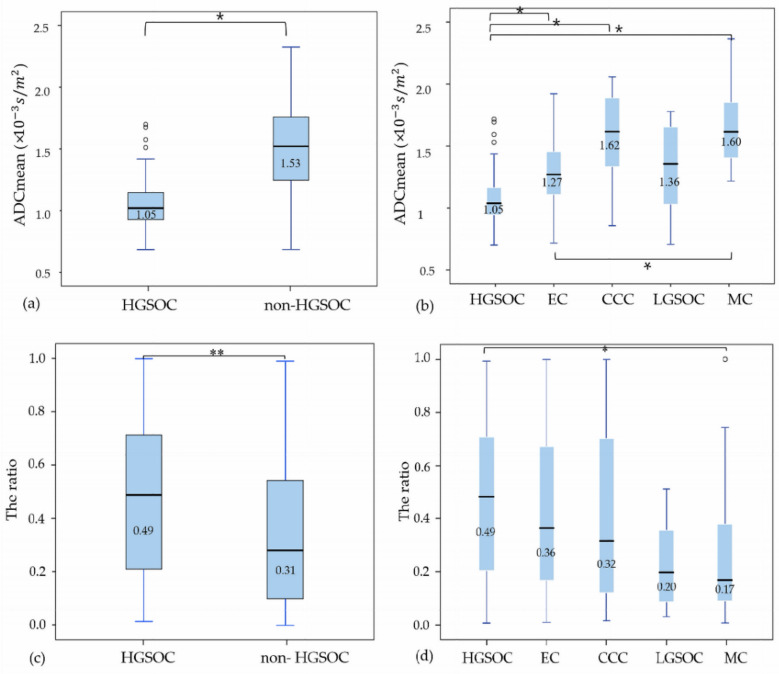
ADC_mean_ and the ratios of the area of the solid portion to the total area of ovarian lesions by histological types of EOC. (**a**) ADC_mean_ between HGSOC and non-HGSOC, (**b**) ADC_mean_ among 5 histological types of EOC, (**c**) the ratio of area of solid portion to total ovarian lesion between HGSOC and non-HGSOC, (**d**) the ratio of area of solid portion to total ovarian lesion among 5 histological types of EOC. * *p* < 0.001; ** *p* = 0.001; EOC: epithelial ovarian cancer, HGSOC: high-grade serous ovarian cancer, EC: endometrioid cancer, CCC: clear-cell carcinoma, LGSOC: low-grade serous ovarian cancer, MC: mucinous carcinoma.

**Figure 4 jcm-11-00229-f004:**
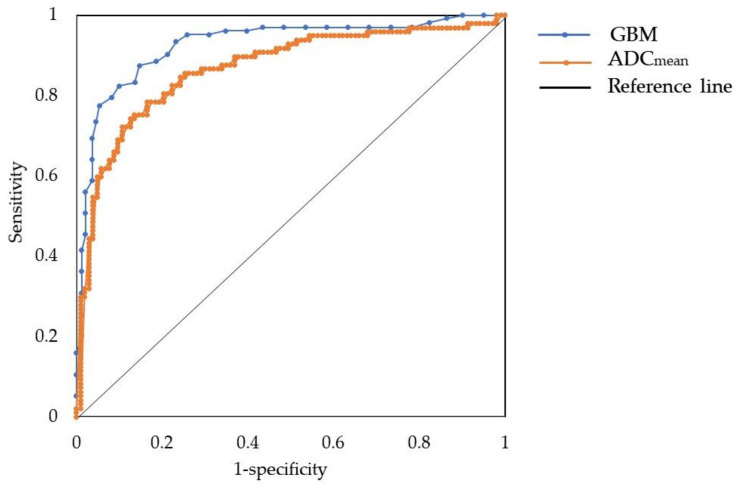
Comparison ROC curve between GBM and logistic regression for ADC_mean_. GBM: Gradient boosting model, ROC: receiver operating characteristic, ADC_mean_: mean value of apparent diffusion coefficient.

**Figure 5 jcm-11-00229-f005:**
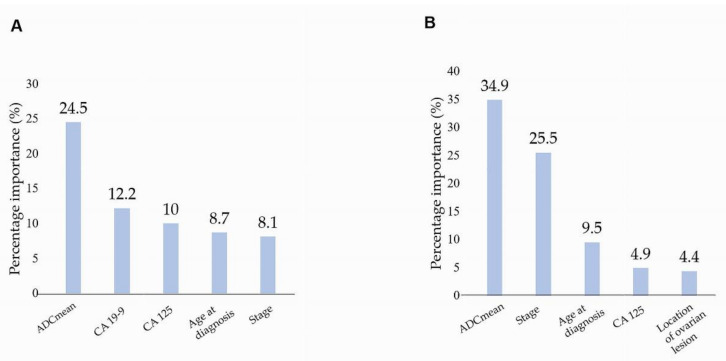
Variable importance by percentage in predicting the outcome. (**A**) Between HGSOC and non-HGSOC, (**B**) among 5 histological types of EOC (HGSOC, EC, CCC, LGSOC, MC). ADCmean: mean value of apparent diffusion coefficient, EOC: epithelial ovarian cancer, HGSOC: high-grade serous ovarian cancer, EC: endometrioid cancer, CCC: clear-cell carcinoma, LGSOC: low-grade serous ovarian cancer, MC: mucinous carcinoma.

**Table 1 jcm-11-00229-t001:** Characteristics of the study population (*n* = 200).

	HGSOC	Non-HGSOC	
	(*n* = 103)	(*n* = 97)	*p* Value
Age at diagnosis (years)	55 (47,61) ^1^	50 (41,60) ^1^	<0.001
Overall survival (months)	47 (13.5,83) ^1^	39 (17,76) ^1^	0.552
Stage	3 (2,3) ^1^	1 (1,2) ^1^	<0.001
I	10 (9.7%)	65 (67.0%)	
II	16 (15.5%)	16 (16.5%)	
III	58 (56.3%)	15 (15.5%)	
IV	19 (18.5)	0 (0%)	
CA 125 (IU/mL)	671.8 (154,2140) ^1^	58.8 (11.9,199) ^1^	<0.001
≤35	8 (7.8%)	38 (39.2%)	
>35	93 (90.3%)	47 (48.5%)	
CA 19-9 (IU/mL)	8.4 (5.0,17.2) ^1^	32 (11.5,94.2) ^1^	<0.001
≤37	74 (71.8%)	37 (38.0%)	
>37	7 (6.8%)	30 (30.9%)	
ADC_mean_ ($\times 10^{-3}\;s/m^{2}$)	1.05 (0.93,1.15) ^1^	1.53 (1.27,1.76) ^1^	<0.001
Total diameter (cm)	7.8 (5.8,9.5) ^1^	11 (8.1,13.2) ^1^	<0.001
Ratio (solid/total cyst, area)	0.46 (0.21,0.67) ^1^	0.31 (0.095,0.525) ^1^	0.001
Location of ovarian lesion			<0.001
Both	42 (41%)	12 (12%)	
Right	32 (31%)	49 (51%)	
Left	29 (28%)	36 (37%)	
Peritoneal seeding			<0.001
Present	66 (64%)	24 (25%)	
Absent	37 (34%)	73 (75%)	

^1^ Median (25th percentile, 75th percentile), HGSOC: high-grade serous ovarian cancer, ADC_mean_: mean value of the apparent diffusion coefficient.

**Table 2 jcm-11-00229-t002:** Multivariate logistic regression of HGSOC.

	Odd Ratio	*p* Value	95% Confidential Interval [CI]
Age at diagnosis	1.048	0.063	[0.998, 1.102]
ADC_mean_ ($\times 10^{-3}\;s/m^{2}$)	0.996	0.001	[0.994, 0.998]
CA 19-9 (IU/mL)	0.973	0.021	[0.951, 0.996]
Stage			
I (reference)			
II	4.16	0.058	[0.954, 18.155]
III	5.98	0.007	[1.626, 21.965]
IV	$1.63\;\times \;10^{9}$	0.98	[0.000]

Reference group: non-high-grade serous ovarian cancer, Nagelkerke $R^{2}$ *n* = 0.732, *p* < 0.001; HGSOC: high-grade serous ovarian cancer, ADC_mean_: mean value of apparent diffusion coefficient.

**Table 3 jcm-11-00229-t003:** Performance metrics of the highest-accuracy machine learning models.

Model	Accuracy	AUC	AUCPR
RF + (HGSOC), (EC), (CC), (LGSOC), (MC)	0.65	0.82	0.63
RF + (HGSOC), (non-HGSOC)	0.89	0.91	0.9
XGBoost + (HGSOC), (EC), (CC), (LGSOC), (MC)	0.68	0.83	0.64
XGBoost + (HGSOC), (non-HGSOC)	0.9	0.92	0.91
GBM + (HGSOC), (EC), (CC), (LGSOC), (MC)	0.67	0.8	0.61
GBM + (HGSOC), (non-HGSOC)	0.91	0.93	0.91

RF: random forest, GBM: gradient boosting machine, XGBoost: extreme gradient boosting machine, AUC: area under the curve, AUCPR: area under the precision recall curve, HGSOC: high-grade serous ovarian cancer, EC: endometrioid cancer, CCC: clear-cell carcinoma, LGSOC: low-grade serous ovarian cancer, MC: mucinous carcinoma.

## Data Availability

Data sharing is not applicable.

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
