# Peer review of "An Application of Machine Learning That Uses the Magnetic Resonance Imaging Metric, Mean Apparent Diffusion Coefficient, to Differentiate between the Histological Types of Ovarian Cancer"

_jcm, 2021, doi:10.3390/jcm11010229_

Round 1
Reviewer 1 Report
Thank you for the opportunity to review this work. The manuscript under this review studies patients with EOC, using a ML to perform a preoperative analysis of the impedance imaging characteristics of images and the clinical factors of patients to predict the histological types of their EOCs.
The study present is interesting, original and novelty.
The presentation of all sections, figures and tables of results are clear.
Author Response
|
Reviewer 1 Thank you for the opportunity to review this work. The manuscript under this review studies patients with EOC, using a ML to perform a preoperative analysis of the impedance imaging characteristics of images and the clinical factors of patients to predict the histological types of their EOCs. The study present is interesting, original and novelty. The presentation of all sections, figures and tables of results are clear. |
We greatly appreciate your constructive comments and suggestions. In this revision, we have improved the paper by addressing the remaining issues raised by the review team. We sincerely thank you for giving us another opportunity to revise the paper. We proofread all manuscript, again to improve readability. We hope that we have addressed the review team’s concerns adequately.
Reviewer 2 Report
This is a well conducted study evaluating the role of preoperative pelvic MRI to differentiate the histological type of EOC.
Despite the authors' statement, I cannot understand the significance of this modality preoperatively at least for the patients with advanced disease. All these patients should be offered standard TAH+BSO plus omentectomy plus pelvic lymphadenectomy. Also for this subset of patients that primary debulking surgery is not feasible, guidelines suggest laparoscopic evaluation and histological confirmation of the disease as a SoC as in accordance to what shown also in this paper imaging modalities could miss more than 20% of diagnoses.
Discrimination between different histological types could be clinically meaningful in early EOC patients. In that case, authors should limit their analysis to stage I tumors and also include borderline cases as in previous studies as borderline tumors and low grade tumors belong to the same continuum of the disease.
Minor comments: Table 1 should be more details, including number of cases per stage. Also, authors should explain why statistical important factors in the univariate analysis were not included in their regression model.
Reviewer 3 Report
Thank you for asking me to review the manuscript entitled “An application of machine learning that uses the magnetic resonance imaging metric, mean apparent diffusion coefficient, to differentiate between the histological types of ovarian cancer” by Heekyoung Song , Seongeun Bak , Imhyeon Kim, Jae Yeon Woo, Eui Jin Cho, Youn Jin Choi, Sung Eun Rha, Shin Ah Oh, Seo Yeon Youn and Sung Jong Lee.
This retrospective single-center study included patients diagnosed with epithelial ovarian cancer with preoperative pelvic magnetic resonance imaging (MRI). The aim of the study is to use machine learning to improve the preoperative differentiation of histologic subtypes of ovarian cancer.
The authors have included 200 patients during the study period (m January 2009 to December 2021) and all of these have had histologic confirmation of the diagnosis: Can you please review the dates? The manuscript was submitted before the availability of histologic results from December 2021.
The methodology describes that the ROI was not drawn independently. The authors should perform a subanalysis on reproducibility on a subset of patients to validate their methodology.
Can the authors comment on the differences between non-HGSC subtypes?
The authors have performed an ROC analysis. Can you please provide Sensitivity, Spec, PPV and NPV for differentiating HGSOC from non-HGSOC.
Author Response
|
Reviewer 3 This retrospective single-center study included patients diagnosed with epithelial ovarian cancer with preoperative pelvic magnetic resonance imaging (MRI). The aim of the study is to use machine learning to improve the preoperative differentiation of histologic subtypes of ovarian cancer. The authors have included 200 patients during the study period (m January 2009 to December 2021) and all of these have had histologic confirmation of the diagnosis: Can you please review the dates? The manuscript was submitted before the availability of histologic results from December 2021. The methodology describes that the ROI was not drawn independently. The authors should perform a subanalysis on reproducibility on a subset of patients to validate their methodology. Can the authors comment on the differences between non-HGSC subtypes? The authors have performed an ROC analysis. Can you please provide Sensitivity, Spec, PPV and NPV for differentiating HGSOC from non-HGSOC. |
Comment 1: The authors have included 200 patients during the study period (m January 2009 to December 2021) and all of these have had histologic confirmation of the diagnosis: Can you please review the dates? The manuscript was submitted before the availability of histologic results from December 2021.
Response 1: Thank you very much for kind advice and we deeply apologize for the mistake in the date. The original date was “January 2009 and December 2020.” The date was changed and highlighted with green block in the revised manuscript. Only patients who were proved pathologically epithelial ovarian cancer were included in this study. In addition, we uploaded IRB approval and surgery date of patients for this study through supplementary materials (raw data). Please review these papers and we’re really sorry about our mistake, again.
Comment 2: The methodology describes that the ROI was not drawn independently. The authors should perform a subanalysis on reproducibility on a subset of patients to validate their methodology.
Response 2: We focused on the possibility for finding solid portion in ADC map by the gynaecologist, therefore, we selected the method, original method that the gynecologist found the lesion in first time and next the radiologist reviewed same lesion. However, we understood the importance of reproducibility this methology as your comment, the radiologist checked ADCmean, the area and diameter of solid portion. So we added below sentence in discussion. “However, it was found that the ICC between the two reviewers (0.966) was excellent.” These contents was added in revised manuscript.
Commnet 3: Can the authors comment on the differences between non-HGSC subtypes?
Response 3: Thank you for giving us great suggestions and we added more descriptions about non-HGSOC. “Additionally, among patients with non-HGSOC, the ADCmean value of patients with EC was the lowest and increased in the order of LGSOC, MC, and CCC. The only significant difference was noted between EC and MC (p<0.001). In addition, EC had the highest ratio of the area of the solid portion to the total area of ovarian lesions, and the ratios of CCC, LGSOC, and MC decreased in non-HGSOC; however, the difference was not significant.”
“ The relationship between the mean ADC value and cellularity in ECs or CCCs is con-troversial. Tanaka et al. reported that CCC had a higher proportion of solid components than EC. On the contrary, we found that CCC had a lower proportions of solid compo-nents than EC, similar to the results of Ono et al. [23,29]. According to the World Health Organization, CCC is histologically defined as a neoplasm composed of clear cells grow-ing in a solid, tubular, or papillary pattern, with hobnail cells lining the tubules and cysts [30]. However, correct histopathological diagnosis can be challenging due to the rarity of CCCs, which have a wide range of histological features [31]. This histological diversity can be used to derive the difference in the mean ADC values for CCC.”
“The ratio of the area of the solid portion to the total area of ovarian lesions also differed between the histological types of EOC. The ratio was noted to be significantly different between HGSOC and non-HGSOC, HGSOC, and MC.”
We inserted this sentences in revised manuscript.
Comment 4: The authors have performed an ROC analysis. Can you please provide Sensitivity, Spec, PPV and NPV for differentiating HGSOC from non-HGSOC.
Response 4: Thank you very much for the helpful comment. We described Sensitivity, Specificity, PPV and NPV for differentiating HGSOC from non-HGSOC by logistic regression analysis in Result 3.4 Logistic regression analysis, page 7. However, the expression was not clear for editors to check detail. We added this sentence in front of page 7, “Logistic regression analysis was conducted between HGSOC and non-HGSOC” and changed the sentence, “The ADCmean value showed the highest AUC by logistic regression analysis, and detailed findings showed that the cut-off value was 1.26 × 10-3 s/m2, with sensitivity and specificity as 0.85 and 0.77, respectively, and a positive predictive value and negative predictive value of 0.79 and 0.83, respectively.”
Response 5: We proofread all manuscript, again. Please check sentences and expression of this manuscript. All we revised for your comments highlighted green block in manuscript. We hope that we have addressed the review team’s concerns adequately.
Round 2
Reviewer 3 Report
The authors have addressed the suggestions raised by the reviewer. Although the study still has limitations - the authors have addressed most of these.